# SLAYER: Spike Layer Error Reassignment in Time

**Sumit Bam Shrestha**[*]
Temasek Laboratories @ NUS
National University of Singapore
Singapore, 117411
tslsbs@nus.edu.sg

**Garrick Orchard**[†]
Temasek Laboratories @ NUS
National University of Singapore
Singapore, 117411
tslgmo@nus.edu.sg

## Abstract

Configuring deep Spiking Neural Networks (SNNs) is an exciting research avenue for low power spike event based computation. However, the spike generation function is non-differentiable and therefore not directly compatible with the standard error backpropagation algorithm. In this paper, we introduce a new general backpropagation mechanism for learning synaptic weights and axonal delays which overcomes the problem of non-differentiability of the spike function and uses a temporal credit assignment policy for backpropagating error to preceding layers. We describe and release a GPU accelerated software implementation of our method which allows training both fully connected and convolutional neural network (CNN) architectures. Using our software, we compare our method against existing SNN based learning approaches and standard ANN to SNN conversion techniques and show that our method achieves state of the art performance for an SNN on the MNIST, NMNIST, DVS Gesture, and TIDIGITS datasets.

## 1   Introduction

Artificial Neural Networks (ANNs), especially Deep Neural Networks, have become the go-to tool for many machine learning tasks. ANNs achieve state of the art performance in applications ranging from image classification and object recognition, to object tracking, signal processing, natural language processing, self driving cars, health care diagnostics, and many more. In the currently popular second generation of ANNs, backpropagation of the error signal to the neurons in preceding layer is the key to their learning prowess.

However, ANNs generally require powerful GPUs and computing clusters to crunch their inputs into useful outputs. Therefore, in scenarios where power consumption is constrained, on-site use of ANNs may not be a viable option. On the other hand, biologically inspired spiking neurons have long shown great theoretical potential as efficient computational units [1–3] and recent advances in Spiking Neural Network (SNN) hardware [4–6] have renewed research interest in this area.

SNNs are similar to ANNs in terms of network topology, but differ in the choice of neuron model. Spiking neurons have memory and use a non-differentiable spiking neuron model (*spike function*) while ANNs typically have no memory and model each neuron using a continuously differentiable activation function. Since the spike function is non-differentiable, the backpropagation mechanism used to train ANNs cannot be directly applied.

Nevertheless, a handful of supervised learning algorithms for SNNs have been proposed previously. The majority of them are designed for a single neuron [7–9], but a few have proposed methods to work around the non-differentiable spike function and backpropagate error through multiple layers [10–14].

---

[*]bam_sumit@hotmail.com

[†]www.garrickorchard.com, garrickorchard@gmail.com

Event based methods such as SpikeProp [10] and EvSpikeProp [11] have the derivative term defined only around the firing time, whereas [12–14] ignore the temporal effect of spike signal. In Section 3.1 we describe the strengths and weaknesses of these approaches in more detail.

The main contribution of this paper is a general method of error backpropagation for SNNs (Section 3) which we call Spike LAYer Error Reassignment (SLAYER). SLAYER distributes the credit of error back through the SNN layers, much like the traditional backprop algorithm distributes error back through an ANN's layers. However, unlike backprop, SLAYER also distributes the credit of error back in time because a spiking neuron's current state depends on its previous states (and therefore, on the previous states of its input neurons). SLAYER can simultaneously learn both synaptic weights and axonal delays, which only a few previous works have attempted [15, 16].

We have developed and released[3] a CUDA accelerated framework to train SNNs using SLAYER. We demonstrate SLAYER achieving state of the art accuracy for an SNN on neuromorphic datasets (Section 4) for visual digit recognition, action recognition, and spoken digit recognition.

The rest of the paper is organized as follows. We start by introducing notation for a general model of a spiking neuron and extending it to a multi-layer SNN in Section 2. Then, in Section 3 we discuss previously published methods for learning SNN parameters before deriving the SLAYER backpropagation formulae. In Section 4, we demonstrate the effectiveness of SLAYER on different benchmark datasets before concluding in Section 5.

## 2 Spiking Neural Network: Background

An SNN is a type of ANN that uses more biologically realistic spiking neurons, as its computational units. In this Section we introduce a model for a spiking neuron before extending the formulation to a multi-layer network of spiking neurons (SNN).

### 2.1 Spiking Neuron Model

Spiking neurons obtain their name from the fact that they only communicate using voltage spikes. All inputs and outputs to the neuron are in the form of spikes, but the neuron maintains an internal state over time. In this paper, we will use a simple yet versatile spiking neuron model known as the Spike Response Model (SRM) [17], described below.

Consider an input spike train to a neuron, $s_i(t) = \sum_f \delta(t - t_i^{(f)})$. Here $t_i^{(f)}$ is the time of the $f^{\text{th}}$ spike of the $i^{\text{th}}$ input. In SRM, the incoming spikes are converted into a *spike response signal*, $a_i(t)$, by convolving $s_i(t)$ with a spike response kernel $\varepsilon(\cdot)$. This can be written as $a_i(t) = (\varepsilon * s_i)(t)$. Similarly, the refractory response of a neuron is represented as $(\nu * s)(t)$, where $\nu(\cdot)$ is the refractory kernel and $s(t)$ is the neuron's output spike train.

Each spike response signal is scaled by a synaptic weight $w_i$ to generate a *Post Synaptic Potential (PSP)*. The neuron's state (membrane potential), $u(t)$, is simply the sum of all PSPs and refractory responses

$$u(t) = \sum_i w_i (\varepsilon * s_i)(t) + (\nu * s)(t) = \boldsymbol{w}^\top \boldsymbol{a}(t) + (\nu * s)(t). \tag{1}$$

An output spike is generated whenever $u(t)$ reaches a predefined threshold $\vartheta$. More formally, the *spike function* $f_s(\cdot)$ is defined as

$$f_s(u) : u \to s, \ \ s(t) := s(t) + \delta(t - t^{(f+1)}) \text{ where } t^{(f+1)} = \min\{t : u(t) = \vartheta, \ t > t^{(f)}\}. \tag{2}$$

Unlike the activation functions used in non-spiking ANNs, the derivative of the spike function is undefined which is a major obstacle for backpropagating error from output to input for SNNs. Also, note that the effect of an input spike is distributed in future via the spike response kernels which is the reason for temporal dependency in the spiking neuron.

The above formulation can be extended to include axonal delays by redefining the spike response kernel as $\varepsilon_d(t) = \varepsilon(t - d)$, where $d \geq 0$ is the axonal delay.[4]

## 2.2 SNN Model

Here we describe a feedforward neural network architecture with $n_l$ layers. This formulation applies to fully connected, convolutional, as well as pooling layers. For implementation details, refer to supplementary material. Consider a layer $l$ with $N_l$ neurons, weights $\boldsymbol{W}^{(l)} = [\boldsymbol{w}_1, \cdots, \boldsymbol{w}_{N_{l+1}}]^\top \in \mathbb{R}^{N_{l+1} \times N_l}$ and axonal delays $\boldsymbol{d}^{(l)} \in \mathbb{R}^{N_l}$. Then the network forward propagation is as described below.

$$\boldsymbol{a}^{(l)}(t) = (\boldsymbol{\varepsilon}_d * \boldsymbol{s}^{(l)})(t) \tag{3}$$

$$\boldsymbol{u}^{(l+1)}(t) = \boldsymbol{W}^{(l)} \boldsymbol{a}^{(l)}(t) + (\nu * \boldsymbol{s}^{(l+1)})(t) \tag{4}$$

$$\boldsymbol{s}^{(l+1)}(t) = f_s(\boldsymbol{u}^{(l+1)}(t)) \tag{5}$$

Also note that the inputs, $\boldsymbol{s}^{(0)}(t) = \boldsymbol{s}^{\text{in}}(t)$, and outputs, $\boldsymbol{s}^{\text{out}}(t) = \boldsymbol{s}^{(n_l)}(t)$, are spike trains rather than numeric values.

# 3 Backpropagation in SNN

In this Section, we first discuss prior works on learning in SNNs before presenting the details of error backpropagation using SLAYER.

## 3.1 Existing Methods

Previous works which use learning to configure a deep SNN (multiple hidden layers) can be grouped into three main categories. The first category uses an ANN to train an equivalent *shadow network*. The other two categories train directly on the SNN but differ in how they approximate the derivative of the spike function.

The first category leverages learning methods for conventional ANNs by training an ANN and converting it to an SNN [18–25] with some loss of accuracy. There are different approaches to overcome the loss of accuracy such as introducing extra constraints on neuron firing rate [23], scaling the weights [23–25], constraining the network parameters [20], formulating an equivalent transfer function for a spiking neuron [19–22], adding noise in the model [21, 22], using probabilistic weights [18] and so on.

The second category keeps track of the membrane potential of spiking neurons only at spike times and backpropagates errors based only on membrane potentials at spike times. Examples include SpikeProp [10] and its derivatives [11, 26]. These methods are prone to the "dead neuron" problem: when no neurons spike, no learning occurs. Heuristic measures are required to revive the network from such a condition.

The third category of methods backpropagate errors based on the membrane potential of a spiking neuron at a single time step only. Different methods are used to approximate the derivative of the spike function. Panda et al. [12] use an expression similar to that of a multi-layer perceptron system, Lee et al. [13] use small signal approximation at the spike times, and Zenke et al. [14] simply propose a surrogate function to serve as the derivative. All these methods ignore the temporal dependency between spikes. They credit the error at a given time step to the input signals at that time step only, thus neglecting the effect of earlier spike inputs.

## 3.2 Backpropagation using SLAYER

In this Section we describe the Loss Function (Section 3.2.1), how error is assigned to previous time-points (Section 3.2.2), and how the derivative of the spike function is approximated (Section 3.2.4).

### 3.2.1 The Loss Function

Consider a loss function for the network in time interval $t \in [0, T]$, defined as

$$E := \int_0^T L(\boldsymbol{s}^{(n_l)}(t), \hat{\boldsymbol{s}}(t)) \, \mathrm{d}t = \frac{1}{2} \int_0^T \left( \boldsymbol{e}^{(n_l)}(\boldsymbol{s}^{(n_l)}(t), \hat{\boldsymbol{s}}(t)) \right)^2 \mathrm{d}t \tag{6}$$

where $\hat{s}(t)$ is the target spike train, $L(s^{(n_l)}(t), \hat{s}(t))$ is the loss at time instance $t$ and $e^{(n_l)}(s^{(n_l)}(t), \hat{s}(t))$ is the error signal at the final layer. For brevity we will write the error signal as $e^{(n_l)}(t)$ from here on.

To learn a target spike train $\hat{s}(t)$ an error signal of the form

$$e^{(n_l)}(t) := \varepsilon * \left( s^{(n_l)}(t) - \hat{s}(t) \right) = a^{(n_l)}(t) - \hat{a}(t) \qquad (7)$$

is a suitable choice. This loss function is similar to the van-Rossum distance [27].

For classification tasks, a decision is typically made based on the number of output spikes during an interval rather than the precise timing of the spikes. To handle such cases, the error signal during the interval can be defined as

$$e^{(n_l)}(t) := \left( \int_{T_{\text{int}}} s^{(n_l)}(\tau)\,\mathrm{d}\tau - \int_{T_{\text{int}}} \hat{s}(\tau)\,\mathrm{d}\tau \right), \qquad t \in T_{\text{int}} \qquad (8)$$

and zero outside the interval $T_{\text{int}}$. Here we only need to define the number of desired spikes during the interval (the second integral term). The actual spike train $\hat{s}(t)$ need not be defined.

### 3.2.2 Temporal Dependencies to History

In the mapping from input spikes, $s^{(l)}(t)$, to membrane potential, $u^{(l+1)}(t)$, temporal dependencies are introduced due to spike response kernel $\varepsilon(\cdot)$ which distributes the effect of input spikes into future time values i.e. the signal $u^{(l+1)}(t)$ is dependent on current as well as past values of inputs $s^{(l)}(t), t \le t_1$. Step based learning approaches [12–14] ignore this temporal dependency and only use signal values at the current time instance. Below we describe how SLAYER accounts for this temporal dependency. Full details of the derivation are provided in the supplementary material.

Let us, for the time being, discretize the system with a sampling time $T_s$ such that $t = n\,T_s, n \in \mathbb{Z}$ and use $N_s$ to denote the total number of samples in the period $t \in [0, T]$. The signal values $a^{(l)}[n]$ and $u^{(l)}[n]$ have a contribution to future network losses at samples $m = n, n+1, \cdots, N_s$. Taking into account the temporal dependency, the gradient term is given by

$$\nabla_{w_i^{(l)}} E := T_s \sum_{n=0}^{N_s} \frac{\partial L[n]}{\partial w_i^{(l)}} = T_s \sum_{m=0}^{N_s} a^{(l)}[m] \sum_{n=m}^{N_s} \frac{\partial L[n]}{\partial u_i^{(l+1)}[m]}. \qquad (9)$$

The backpropagation estimate of error in layer $l$ is then

$$e^{(l)}[n] := \sum_{m=n}^{N_s} \frac{\partial L[m]}{\partial a^{(l)}[n]} = \left( W^{(l)} \right)^\top \delta^{(l+1)}[n] \qquad (10)$$

$$\delta^{(l)}[n] := \sum_{m=n}^{N_s} \frac{\partial L[m]}{\partial u^{(l)}[n]} = f_s'(u^{(l)}[n]) \cdot \left( \varepsilon_d \odot e^{(l)} \right)[n]. \qquad (11)$$

Here $\odot$ represents element-wise correlation operation in time. The summation from $n$ to $N_s$ assigns the credit of all the network losses in a future time to the neuron at current time. Note that at the output layer, $\partial L[m]/\partial a^{(n_l)}[n] = 0$ for $n \ne m$ which results in $e^{(n_l)}[n] = \partial L[n]/\partial a^{(n_l)}[n]$. This is in agreement with the definition of output layer error in (6).

Similarly for axonal delay with $\dot{a}^{(l)} = \left( \dot{\varepsilon}_d * s^{(l)} \right)$, one can derive the delay gradient as follows.

$$\nabla_{d^{(l)}} E := T_s \sum_{n=0}^{N_s} \frac{\partial L[n]}{\partial d^{(l)}} = -T_s \sum_{m=0}^{N_s} \dot{a}^{(l)}[m] \cdot e^{(l)}[m] \qquad (12)$$

### 3.2.3 The Derivative of the Spike Function

The derivative of the spike function is always a problem for supervised learning in a multilayer SNN. In Section 3.1, we discussed how prior works handle the derivative. Below we describe how SLAYER deals with the spike function derivative.

Consider the state of a spiking neuron at time $t = \tau$. The neuron can either be in spiking state $(u(\tau) \ge \vartheta)$ or non-spiking state $(u(\tau) < \vartheta)$. Now consider a perturbation in the membrane potential by an amount $\Delta u(\tau) = \pm \Delta\zeta$ for $\Delta\zeta > 0$.

A neuron in the non-spiking state will switch to the spiking state when perturbed by $+\Delta\zeta$ if $+\Delta\zeta \ge \vartheta - u(\tau)$. Similarly, a neuron in the spiking state will switch to the non-spiking state when

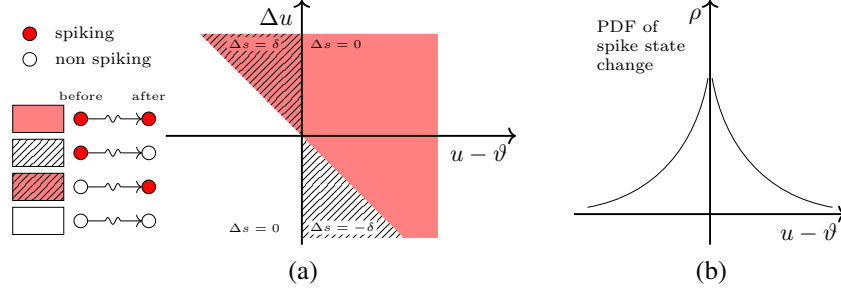

Figure 1: (a) Transition of a spiking neuron's state due to random perturbation $\Delta\zeta$. (b) Probability density function of spike state change.

perturbed by $-\Delta\zeta$ if $-\Delta\zeta < \vartheta - u(\tau)$. In both the cases, when there is a change in spiking state of the neuron when $\Delta\zeta > |u(\tau) - \vartheta|$. Fig. 1(a) shows these transitions. Therefore,

$$\frac{\Delta s(\tau)}{\Delta u(\tau)} = \begin{cases} \frac{\delta(t-\tau)}{\Delta\zeta} & \text{when } \Delta\zeta > |u(\tau) - \vartheta| \\ 0 & \text{otherwise} \end{cases}. \tag{13}$$

This formulation is still problematic because of Dirac-delta function. However, we can see that the derivative term is biased towards zero as $|u(\tau) - \vartheta|$ increases. A good estimate of the derivative term $f'_s(\cdot)$ can be made using the probability of a change in spiking state.

If we denote the probability density function as $\rho(t)$, then the probability of spiking state change in an infinitesimal time window of width $\Delta t$ around $\tau$ and a small perturbation $\Delta\zeta \to 0$ as $\Delta u \to 0$ can be written as $\rho(\tau)\,\Delta\zeta\,\Delta t$. Now, the expected value of $f'_s(\tau)$ can be written as

$$E[f'_s(\tau)] = \lim_{\substack{\Delta\zeta \to 0 \\ \Delta t \to 0}} \left( \rho(\tau)\,\Delta\zeta\,\Delta t \frac{1}{\Delta t\,\Delta\zeta} + (1 - \rho(\tau)\,\Delta\zeta\,\Delta t) \times 0 \right) = \rho(\tau). \tag{14}$$

The derivative of spike function represents the Probability Density Function (PDF) for change of state of a spiking neuron. For a completely deterministic spiking neuron model, it is a sum of impulses at spike times, which is equivalent to the spike train $s(t)$. Nevertheless, we can relax the deterministic nature of spiking neuron and use the stochastic spiking neuron approximation for backpropagating errors.

The function $\rho(t) = \rho(u(t) - \vartheta)$ must be high when $u(\tau)$ is close to $\vartheta$ and must decrease as it moves further away. An example PDF is shown in Figure 1(b). A good formulation of this function is the spike escape rate function [28, 29] $\rho(t)$ which is usually represented by an exponentially decaying function of $\vartheta - u(\tau)$

$$\rho(t) = \alpha \exp(-\beta\,|u(t) - \vartheta|). \tag{15}$$

The negative portion of fast sigmoid function, used in [14], is also a suitable candidate for $\rho(u(t) - \vartheta)$.

### 3.2.4 The SLAYER Backpropagation Pipeline

Now, applying the limit $T_s \to 0$ for (9) (10) and (11) and using the expectation value of $f'_s(t)$, we arrive at the SLAYER backpropagation pipeline.

$$\boldsymbol{e}^{(l)}(t) = \begin{cases} \frac{\partial L(t)}{\partial \boldsymbol{a}^{(n_l)}} & \text{if } l = n_l \\ \left(\boldsymbol{W}^{(l)}\right)^{\top} \boldsymbol{\delta}^{(l+1)}(t) & \text{otherwise} \end{cases} \tag{16}$$

$$\boldsymbol{\delta}^{(l)}(t) = \boldsymbol{\rho}^{(l)}(t) \cdot \left(\boldsymbol{\varepsilon}_d \odot \boldsymbol{e}^{(l)}\right)(t) \tag{17}$$

$$\nabla_{\boldsymbol{W}^{(l)}} E = \int_0^T \boldsymbol{\delta}^{(l+1)}(t) \left(\boldsymbol{a}^{(l)}(t)\right)^{\top} \mathrm{d}t \tag{18}$$

$$\nabla_{\boldsymbol{d}^{(l)}} E = -\int_0^T \dot{\boldsymbol{a}}^{(l)}(t) \cdot \boldsymbol{e}^{(l)}(t)\,\mathrm{d}t \tag{19}$$

The gradients with respect to weights and delays are given by (18) and (19). It is straightforward to use any of the optimization techniques from simple gradient descent method to adaptive methods such as RmsProp, ADAM, and NADAM to drive the network towards convergence.

# 4 Experiments and Results

In this Section we will present different experiments conducted and results on them to evaluate the performance of SLAYER. First, we train an SNN to produce a fixed Poisson spike train pattern in response to a given set of Poisson spike inputs. We use this simple example to show how SLAYER works. Afterwards we present results of classification tasks performed on both spiking datasets and non-spiking datasets converted to spikes.

Simulating an SNN is a time consuming process due to the additional temporal dimension of signals. An efficient simulation framework is key to enabling training on practical spiking datasets. We use our CUDA accelerated SNN deep learning framework for SLAYER to perform all the simulations for which results are presented in this paper. All the accuracy values reported for SLAYER are averaged over 5 different independent trials. In our experiments, we use spike response kernels of the form $\varepsilon(t) = {}^t/_{\tau_s} \exp(1 - {}^t/_{\tau_s})\Theta(t)$ and $\nu(t) = -2\vartheta \exp(1 - {}^t/_{\tau_r})\Theta(t)$. Here, $\Theta(t)$ is the Heaviside step function. SLAYER, however, is independent of the choice of the kernels.

Throughout this paper, we will use the following notation to indicate the SNN architecture. Layers are separated by - and spatial dimensions are separated by x. A convolution layer is represented by c and an aggregation layer is represented by a. For example 34x34x2-8c5-2a-5o represents a 4 layer SNN with $32 \times 34 \times 2$ input, followed by 8 convolution filters ($5 \times 5$), followed by $2 \times 2$ aggregation layer and finally a dense layer connected to 5 output neurons.

## 4.1 Poisson Spike Train

This is a simple experiment to help understand the learning process in SLAYER. A Poisson spike train was generated for 250 different inputs over an interval of $50$ ms. Similarly a target spike train was generated using a Poisson distribution. The task is to learn to fire the desired spike train for the random spike inputs using an SNN with 25 hidden neurons.

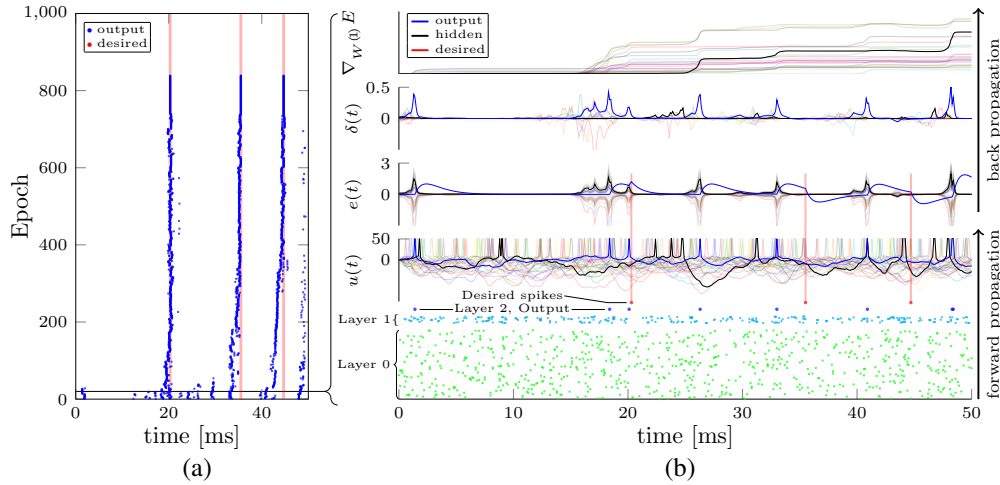

Figure 2: (a) Spike Raster plot during Poisson spike train learning. (b) Snapshot of SLAYER backpropagated learning signals at 20th learning epoch.

The learning plots are shown in Figure 2. From the learning spike raster, we can see that initially there are output spikes distributed at random times (Figure 2(a) bottom). As learning progresses, the unwanted spikes are suppressed and the spikes near the desired spike train are reinforced. The learning finally converges to the desired spike train at the 739th epoch. The learning snapshot at epoch 20 (Figure 2(b)), shows how the error signal is constructed. The spike raster for input, hidden

and output layer is shown at the bottom, The blue plots show the respective signals for output layer. The conversion from error signal, $e$, to delta signal, $\delta$, shows that the error credit assigned depends on the membrane potential value $u$. Note the temporal credit assignment of error. A nonzero value of $e$ results in non-zero values of $\delta$ at earlier points in time, even if the error signal, $e$, was zero at those times. Similar observations can be made for hidden layer signals. Out of 25 hidden layer signals, one is highlighted in black and rest are shown faded.

Table 1: Benchmark Classification Results

| Dataset | Method | Architecture | Accuracy |
|---|---|---|---|
| MNIST | Lee et al. [13] | `28x28-800-10` | $99.31\%$ |
| | Rueckauer et al. [25] | SNN converted from standard ANN | $\mathbf{99.44}\%$ |
| | SLAYER | `28x28-12c5-2a-64c5-2a-10o` | $99.36 \pm 0.05\%$ |
| NMNIST | Lee et al. [13] | `34x34x2-800-10` | $98.66\%$ |
| | SKIM [30] | `34x34x2-10000-10` | $92.87\%$ |
| | DART [31] | DART feature descriptor | $97.95\%$ |
| | SLAYER | `34x34x2-500-500-10` | $98.89 \pm 0.06\%$ |
| | SLAYER | `34x34x2-12c5-2a-64c5-2a-10o` | $\mathbf{99.20 \pm 0.02}\%$ |
| DVS Gesture | TrueNorth [32] | SNN (16 layers) | $91.77\% (\mathbf{94.59}\%)$ |
| | SLAYER | SNN (8 layers) | $\mathbf{93.64 \pm 0.49}\%$ |
| TIDIGITS | SOM-SNN [33] | MFCC-SOM-SNN | $97.6\%$ |
| | Tavanaei et al. [34] | Spiking CNN and HMM | $96.00\%$ |
| | SLAYER | MFCC-SOM, `484-500-500-11` | $\mathbf{99.09 \pm 0.13}\%$ |

## 4.2 MNIST Digit Classification

MNIST is a popular machine learning dataset. The task is to classify an image containing a single digit. This dataset is a standard benchmark to test the performance of a learning algorithm. Since SLAYER is a spike based learning algorithm, the images are converted into spike trains spanning 25 ms using Generalized Integrate and Fire Model of neuron [35]. Standard split of 60,000 training samples and 10,000 testing samples was used with no data augmentation. For classification, we use the spike counting strategy. During training, we specify a target of 20 spikes for the true neuron and 5 spikes for each false neuron over the 25 ms period. During testing, the output class is the class which generates the highest spike count.

The classification accuracy of SLAYER for MNIST classification is listed in Table 1 along with other SNN based approaches. We achieve testing accuracy of $99.36\%$ on the network which is the best result for completely SNN based learning. Although this accuracy does not fare well with state of the art deep learning methods, for an SNN based approach it is a commendable result.

## 4.3 NMNIST Digit Classification

The NMNIST dataset [36] consists of MNIST images converted into a spiking dataset using a Dynamic Vision Sensor (DVS) moving on a pan-tilt unit. Each dataset sample is 300 ms long, and $34{\times}34$ pixels big, containing both 'on' and 'off' spikes. This dataset is harder than MNIST because one has to deal with saccadic motion. For NMNIST training, we use a target of 10 spikes for each false class neuron and 60 spikes for the true class neuron. The output class is the one with greater spike count. The training and testing separation is the same as the standard MNIST split of 60,000 training samples and 10,000 testing samples. The NMNIST data was not stabilized before feeding to the network.

The results on NMNIST classification listed in Table 1 show that SLAYER learning surpasses the current reported state of the art result on NMNIST dataset by Lee et. al. [13] with a comparable number of neurons. However, the CNN architecture trained with SLAYER achieves the best result.

## 4.4 DVS Gesture Classification

The DVS Gesture [32] dataset consists of recordings of 29 different individuals performing 10 different actions such as clapping, hand wave etc. The actions are recorded using a DVS camera

under three different lighting conditions. The problem is to classify the action sequence video into an action label. The dataset allows us to test SLAYER on a temporal task. For training we set a target spike count of 30 for false class neurons and 180 for the true class neuron. Samples from the first 23 subjects were used for training and last 6 subjects were used for testing.

The results for DVS Gesture classification are listed in Table 1. SLAYER achieves a very good testing accuracy of $93.64\%$ on average. In SLAYER training as well as testing, only the first $1.5$ s out of $\approx 6$ s of action video for each class were used to classify the actions. For speed reasons, the SNN was simulated with a temporal resolution of 5 ms. Despite these shortcomings, the accuracy results are excellent, surpassing the testing accuracy of TrueNorth trained with EEDN [32]. With output filtering, the TrueNorth accuracy can be increased to $94.59\%$. Nevertheless, SLAYER is able to classify with a significantly less number of neurons and layers. The TrueNorth approach uses additional neurons before the CNN classifier for pre-processing, whereas in SLAYER, the spike data from the DVS is directly fed into the classifier.

### 4.5 TIDIGITS Classification

TIDIGITS [37] is an audio classification dataset containing audio signals corresponding to digit utterances from 'zero' to 'nine' and 'oh'. In this paper, we use audio data converted to spikes using the MFCC transform followed by a Self Organizing Map (SOM) as described in [33].For training, we specify a target of 5 spikes for false classes and 20 spikes for the true class. The dataset was split into 3950 training samples and 1000 testing samples.

The results for TIDIGITS classification are listed in Table 1. SLAYER significantly improves upon the testing accuracy results of SNN based approach using SOM-SNN [33] on the same encoded spike inputs. However, the best reported accuracy for TIDIGITS classification $99.7\%$ [38] is using MFCC and HMM-GMM approach (non spiking). The accuracy of SLAYER, however, is still competitive at $99.09\%$.

## 5 Discussion

We have proposed a new error backpropagation for SNNs which properly considers the temporal dependency between input and output signals of a spiking neuron, handles the non-differentiable nature of the spike function, and is not prone to the dead neuron problem. The result is SLAYER, a learning algorithm for learning both weight and axonal delay parameters in an SNN. We have demonstrated SLAYER's effectiveness in achieving state of the art accuracy for an SNN on spoken digit and visual digit recognition as well as visual action recognition.

During training, we require both true and false neurons to fire, but specify a much higher spike count target for the true class neuron. This approach prevents neurons from going dormant and they easily learn to fire more frequently again when required. The desired spike count was chosen to be roughly proportional to the simulation interval.

With proper scaling factor of surrogate function (15), vanishing or exploding gradients are not an issue in SLAYER for any of the networks we have trained thus far, and we believe that SLAYER can be used on even deeper networks.

The temporal error credit assignment and axonal delay do increase the computational complexity requirements, respectively comprising $8.03\%$ and $2.55\%$ of the computation time for the training of fully connected NMNIST network, the computational overhead is not significant.

We believe that SLAYER is an important contribution towards efforts to implement backpropagation in an SNN. The development of a CUDA accelerated learning framework for SLAYER was instrumental in tackling bigger datasets in SNN domain, although they are still not big when compared to the huge datasets tackled by conventional (non-spiking) deep learning.

Neuromorphic hardware such as TrueNorth [4], SpiNNaker [5], Intel Loihi [6] show the potential of implementing large spiking neural networks in an extremely low power chip. These chips usually do not have learning mechanism, or have a primitive learning mechanism built into them. Learning must typically be done offline. SLAYER has good potential to serve as an offline training system to configure a network before deploying it to a chip.

## Footnotes

[3] The code for SLAYER learning framework is publicly available at: `https://bitbucket.org/bamsumit/slayer` A brief video description of this work is available at: `https://www.youtube.com/watch?v=JGdatqqci5o`

[4] Synaptic delay can also be modelled in similar manner. Here, we only consider axonal delay for simplicity.

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
