[Supplementary Material]

# SLAYER: Supplementary Material

May 14, 2018

## 1  SNN architectures

Consider a layer $l$ with $N_l$ neurons, weights $\boldsymbol{W}^{(l)} = [\boldsymbol{w}_1, \cdots, \boldsymbol{w}_{N_{l+1}}]^\top \in \mathbb{R}^{N_{l+1} \times N_l}$ and axonal delays $\boldsymbol{d}^{(l)} \in \mathbb{R}^{N_l}$. Then the network forward propagation routine is

$$\boldsymbol{a}^{(l)}(t) = (\boldsymbol{\varepsilon}_d * \boldsymbol{s}^{(l)})(t) \tag{1}$$

$$\boldsymbol{u}^{(l+1)}(t) = \boldsymbol{W}^{(l)}\,\boldsymbol{a}^{(l)}(t) + (\nu * \boldsymbol{s}^{(l+1)})(t) \tag{2}$$

$$\boldsymbol{s}^{(l+1)}(t) = f_s(\boldsymbol{u}^{(l+1)}(t)). \tag{3}$$

This formulation applies to a fully connected layer with a dense weight matrix. For a convolution layer, the accumulation is via convolution weights which is a subset of fully connected weight matrix. A conversion from convolution weights to fully connected weights is also possible, but it is seldom used in practice as convolution operation can be applied very efficiently.

When it comes to max-pooling, the operation is not straightforward. However, since spikes are sparse events, summing operation of incoming spikes is equivalent to max pooling operation at each time step. Therefore a spiking neuron aggregating spikes with weights slightly greater than its membrane threshold, $\vartheta$, is equivalent to max-pooling operation. The weights are chosen to ensure there is an output for each input spike. Due to refractory dynamics, however, closely timed incoming spikes may be lost sometimes.

## 2  SLAYER derivation: Temporal Dependencies to History

Discretize the SLAYER learning framework with a sampling time $T_s$ such that $t = n\,T_s, n \in \mathbb{Z}$ and use $N_s$ to denote the total number of samples in the period $t \in [0, T]$. The signal values $\boldsymbol{a}^{(l)}[n]$ and $\boldsymbol{u}^{(l)}[n]$ have a contribution to future network losses at samples $m = n, n+1, \cdots, N_s$. Taking into account the temporal dependency, the gradient term is given by

$$\nabla_{\boldsymbol{w}_i^{(l)}} E = T_s \sum_{n=0}^{N_s} \frac{\partial L[n]}{\partial \boldsymbol{w}_i^{(l)}}$$

Using chain rule and noting that the loss at $n$ is dependent on all previous values of $u$, we get

$$\nabla_{\boldsymbol{w}_i^{(l)}} E = T_s \sum_{n=0}^{N_s} \sum_{m=0}^{n} \frac{\partial u_i^{(l+1)}[m]}{\partial \boldsymbol{w}_i^{(l)}} \frac{\partial L[n]}{\partial u_i^{(l+1)}[m]}$$

$$= T_s \sum_{n=0}^{N_s} \sum_{m=0}^{n} \boldsymbol{a}^{(l)}[m] \frac{\partial L[n]}{\partial u_i^{(l+1)}[m]}$$

Changing the order of summation, one can obtain

$$\nabla_{\boldsymbol{w}_i^{(l)}} E = T_s \sum_{m=0}^{N_s} \boldsymbol{a}^{(l)}[m] \sum_{n=m}^{N_s} \frac{\partial L[n]}{\partial u_i^{(l+1)}[m]} \tag{4}$$

The backpropagation estimate of error in layer $l$ are defined as follows.

$$\boldsymbol{e}^{(l)}[n] = \sum_{m=n}^{N_s} \frac{\partial L[m]}{\partial \boldsymbol{a}^{(l)}[n]} \tag{5}$$

$$\& \quad \boldsymbol{\delta}^{(l)}[n] = \sum_{m=n}^{N_s} \frac{\partial L[m]}{\partial \boldsymbol{u}^{(l)}[n]} \tag{6}$$

The formulation is similar to that in backpropagation for standard ANN where $\boldsymbol{e}^{(l)} = \frac{\partial L}{\partial \boldsymbol{a}^{(l)}}$ and $\boldsymbol{\delta}^{(l)} = \frac{\partial L}{\partial \boldsymbol{z}^{(l)}}$. However, in SNN there is notion of time. Also an error at current time effects the network cost at future time instances. To incorporate it, we need to include the network loss at current time as well as all the future times.

For error term, using chain rule and expanding along membrane potential variable, we get

$$\boldsymbol{e}^{(l)}[n] = \sum_{m=n}^{N_s} \frac{\partial L[m]}{\partial \boldsymbol{a}^{(l)}[n]}$$

$$= \sum_{m=n}^{N_s} \frac{\partial \boldsymbol{u}^{(l+1)}[n]}{\partial \boldsymbol{a}^{(l)}[n]} \frac{\partial L[m]}{\partial \boldsymbol{u}^{(l+1)}[n]}$$

$$= \left( \boldsymbol{W}^{(l)} \right)^{\top} \boldsymbol{\delta}^{(l+1)}[n]. \tag{7}$$

For the delta term, using chain rule with intermediate variable $\boldsymbol{a}^{(l)}[k]$, we get

$$\boldsymbol{\delta}^{(l)}[n] = \sum_{m=n}^{N_s} \frac{\partial L[m]}{\partial \boldsymbol{u}^{(l)}[n]}$$

$$= \sum_{m=n}^{N_s} \sum_{k=n}^{m} \frac{\partial \boldsymbol{a}^{(l)}[k]}{\partial \boldsymbol{u}^{(l)}[n]} \frac{\partial L[m]}{\partial \boldsymbol{a}^{(l)}[k]}$$

Chainging the order of integration sign, we get

$$\boldsymbol{\delta}^{(l)}[n] = \sum_{k=n}^{N_s} \frac{\partial(\boldsymbol{\varepsilon}_d * \boldsymbol{s}^{(l)})[k]}{\partial \boldsymbol{u}^{(l)}[n]} \sum_{m=k}^{N_s} \frac{\partial L[m]}{\partial \boldsymbol{a}^{(l)}[k]}$$

Note that the relation between $u$ and $s$ is via the spike function only which only takes into account the membrane potential at current time. Further, each spiking neuron is responsible for one mapping from $u$ to $s$, $\frac{\partial \boldsymbol{s}^{(l)}[n]}{\partial \boldsymbol{u}^{(l)}[n]}$ is a diagonal matrix with $f'_s(u_i^{(l)}[n])$ in its diagonals.

$$\begin{aligned}
\boldsymbol{\delta}^{(l)}[n] &= \sum_{k=n}^{N_s} T_s \, \boldsymbol{\varepsilon}_d[k-n] \cdot f'_s(\boldsymbol{u}^{(l)}[n]) \cdot \boldsymbol{e}^{(l)}[k] \\
&= f'_s(\boldsymbol{u}^{(l)}[n]) \cdot \sum_{k=n}^{N_s} T_s \, \boldsymbol{\varepsilon}_d[k-n] \cdot \boldsymbol{e}^{(l)}[k] \\
&= f'_s(\boldsymbol{u}^{(l)}[n]) \cdot \left(\boldsymbol{\varepsilon}_d \odot \boldsymbol{e}^{(l)}\right)[n]
\end{aligned} \tag{8}$$

Similarly for delay gradient,

$$\nabla_{\boldsymbol{d}^{(l)}} E = T_s \sum_{n=0}^{N_s} \frac{\partial L[n]}{\partial \boldsymbol{d}^{(l)}}$$

Using chain rule and noting that the loss at $n$ is dependent on all previous values of $a$, we get

$$\nabla_{\boldsymbol{d}^{(l)}} E = T_s \sum_{n=0}^{N_s} \sum_{m=0}^{n} \frac{\partial \boldsymbol{a}^{(l)}[m]}{\partial \boldsymbol{d}^{(l)}} \frac{\partial L[n]}{\partial \boldsymbol{a}^{(l)}[m]}$$

Denote $\dot{\boldsymbol{a}}^{(l)} = \left(\dot{\boldsymbol{\varepsilon}}_d * \boldsymbol{s}^{(l)}\right)$. Since the delays are for each individual axon, $\frac{\partial \boldsymbol{a}^{(l)}[m]}{\partial \boldsymbol{d}^{(l)}}$ is a diagonal matrix with $\dot{\boldsymbol{a}}^{(l)}$ in diagonal. Therefore,

$$\nabla_{\boldsymbol{d}^{(l)}} E = -T_s \sum_{n=0}^{N_s} \sum_{m=0}^{n} \dot{\boldsymbol{a}}^{(l)}[m] \cdot \frac{\partial L[n]}{\partial \boldsymbol{a}^{(l)}[m]}$$

Changing the order of summation, one can obtain

$$\begin{aligned}
\nabla_{\boldsymbol{d}^{(l)}} E &= -T_s \sum_{m=0}^{N_s} \dot{\boldsymbol{a}}^{(l)}[m] \cdot \sum_{n=m}^{N_s} \frac{\partial L[n]}{\partial \boldsymbol{a}^{(l)}[m]} \\
&= -T_s \sum_{m=0}^{N_s} \dot{\boldsymbol{a}}^{(l)}[m] \cdot \boldsymbol{e}^{(l)}[m]
\end{aligned} \tag{9}$$

# 3 Sample Results

Here we describe videos showing sample results from NMNIST training and DVS Gesture training on random samples drawn from training partition.

## 3.1 NMNIST Results

The video `SLAYER NMNIST.avi` shows the performance of SNN being tested on 12 different random samples from testing set. For each sample, the spike input is shown on the top left. The red pixels indicate on spike and the blue pixel indicate off spike. The corresponding grayscale MNIST image is shown in the top middle. On the top right, a plot of spike count of all ten output neurons for the particular input is shown. The neuron with higher spike count value is the true class. Note that this plot does not indicate the confidence of the network, as the SNN is trained to spike 10 times even for false classes. On the bottom plot, a spike raster showing output spike of the network over the period of input sample is shown. The video is slowed down $10\times$ to easily visualize the decision process being made. The output correctly classifies all the inputs shown in this video.

## 3.2 DVS Gesture Results

The video `SLAYER DVSGesture.avi` shows the performance of SNN being tested on 24 different random samples from testing set. Fore each sample, the spike input is shown on the top left. The red pixels indicate on spike and the blue pixel indicate off spike. On the top right, a plot of spike count of all eleven output neurons for the particular input is shown. On the bottom plot, a spike raster showing output spike of the network over the period of input sample is shown.