[Reviews · NeurIPS 2018]

Reviewer 1



This paper describes a new learning algorithm for training Spiking Neural Networks which supports multiple layers in an analogous format as backpropagation for training deep neural networks. A unique contribution of this work is the ability to perform temporal credit assignment over past spikes from loss incurred at the current timepoint. This contrasts with previous work which focuses on credit assignment at the time point in which the loss is incurred and ignores the impact of earlier spikes. This paper is very well written. Explanation of their current mathematical definition of a spiking neuron to how they stack and finally how to use the SLAYER algorithm to train those multiple layers flows very nicely. There are a few clarifications that I think would improve the clarity of how the error signals are backpropagated through the network: Equations 9-11 would have better clarity for me if it started from the partial derivative terms necessary for computing the update and then describing the chain rule process for each layer (Equations 10 and 11). While it is understandable that the current work on spiking neural networks does not currently compare favorably to artificial neural networks, such comparisons would be scientifically useful to understand the fundamental gaps in performance that still exists between these models. For example, the performance of the ANN counterpart for the convolutional architectures described on the datasets tested would give more information about how well such an architecture could potentially perform. The spiking algorithms they do compare against use different architectures. It would be important to understand how the architecture changes in this work contribute to the observed differences in performance. Furthermore, does this algorithm continue to perform well networks with more than 10 layers? Are there similar improvements as more layers are added in a similar fashion to what has been shown in the deep learning community? Minor Issues: Line 90: The other three categories 'aggregation' layers described is more commonly referred to as 'pooling' Many of the equations (ex: Eq 6,7,8) are definitions and should be annotated as :=

Reviewer 2



This work contributes a new error backpropagation method for spiking neural networks which is capable of learning both weight and axonal delay parameters in a spiking neural network, and establishes a new SOTA on multiple tasks. This work helps address the long-held challenge in the neuromorphic community of designing a learning algorithm capable of learning (approximately) as well as standard deep neural networks for fixed inputs, but is also capable of learning from time-varying signals which more closely match the native representation of spiking neural networks. This submission was prepared carefully and clearly, with a thorough set of comparison benchmarks to other standard spiking benchmarks. The benchmarks further span both image and audio challenges. Recent work is compared against and referenced, and CNNs and fully-connected architectures similar to those used in the state-of-the-art are used. The derivation of the modified backpropagation rule is well-described. Further, a GPU-optimized toolbox will also be released, all bolstering the work's quality, clarity, and significance. Discretizing temporal history to enable backpropagation and approximating a Dirac function with a more easily-differentiable function is not an entirely surprising approach, but it is used effectively here. It would be useful to see the rule tested on more difficult temporal data than TIDIGITS, in which the SOM on top of MFCCs provides significant precomputation; similarly, a more thorough examination of learning axonal delays would improve the motivation for adding the additional complexity. For example, do learned delays in NMNIST correspond to saccade timing, or just play a role in expanding the parameter space for learning? Finally, some of the choices, e.g. spike rates for true vs. negative classes seem arbitrary. Why were 180 spikes/sec used in DVS gesture classification for true and 30 spikes/sec used for false classes? The work could be improved by comparing against ANNs on spiking data, and referencing works such as [1] that featurize on top of neuromorphic data to suggest whether the limited performance is now more from the SNN models or the input streams themselves. However these are small points. Overall, this work has the potential to strongly impact researchers working in neuromorphic models. [1] Anumula, Jithendar, et al. “Feature Representations for Neuromorphic Audio Spike Streams.” Frontiers in Neuroscience 12 (2018): 23. https://doi.org/10.3389/fnins.2018.00023.

Reviewer 3



The authors propose a method of training spiking neural network models inspired by gradient descent via backpropagation. As with similar work, a surrogate function is used in place of partial derivatives through the neuron activation (the spike function). The main contribution of this work is to extend this approach to account for the temporal dependency between spikes --- i.e. the credit for the error signal is distributed to the entire activity history of a neuron, and not just the most recent time step. This seems computationally expensive, and the model is sophisticated, so it is good that the authors are releasing code to go along with the paper; the actual implementation seems non-trivial. The paper details a new approach for training networks that could then be used on efficient neuromorphic hardware. It is not yet clear from the experiments whether the possibility of improving performance is worth the extra computation, but the experiments do demonstrate learning in networks of hundreds of thousands of parameters. In the experiments, how are the updates (Equations 18 and 19) actually computed? The paper is well organized, and the algorithm and experiments are very clearly presented.